# Effect of gravity on colloidal particle transport in a saturated porous medium: Analytical solutions and experiments

**Shujie Tu, Xiaoming Liu** [ORCID] *, **Hongjiang Cai**

College of Civil Engineering, Hunan University, Changsha, China

* 1478653606@qq.com

**Data Availability Statement:** All relevant data are within the manuscript and its Supporting Information files.

**Funding:** This research was funded by the National Key R&D Program of China (2019YFC1904705),

## Abstract

The colloidal particle transport process in all porous media from laboratory to nature is affected by gravity. In this paper, a mathematical model of colloidal particle migration in a saturated porous medium with the gravity effect is established by combining the gap velocity (advection) with the settling velocity (gravity effect), and an analytical solution of the particle migration problem with time variation of the particle injection intensity is obtained using an integral transformation. The correctness and rationality of the analytical solution are verified by comparing the experimental and theoretical results of the particle migration problem in the point-source transient injection mode. The analytical solution can easily analyze the colloid transport experimental data in a variety of seepage directions. Analysis of the influence of seepage velocities in three different seepage directions on particle transport parameters shows: under the same seepage direction, the peak value of the breakthrough curve increased with an increase in the seepage velocity. The dispersion, adsorption coefficient, and deposition rate decreased with an increase in the seepage velocity. Under the same seepage velocity, the peak value of the breakthrough curve from large to small was vertically downward (VD)> horizontal (H)> vertically upward (VU), the order of dispersion from large to small was vertically downward (VD)>horizontal (H) >vertically upward (VU), the order of the adsorption coefficient and deposition rate of particles from large to small was vertically upward (VU)> horizontal (H) >vertically downward (VD), and the smaller the seepage velocity, the greater the relative differences in the peak value of the breakthrough curve, dispersion, the particle adsorption coefficient, and the deposition rate in the different seepage directions. Therefore, gravity is an important mechanism of particle migration in saturated porous media. The larger the particle size and density were, the smaller the seepage velocity was and the more obvious the effect of gravity. The findings of this study can help for better understanding of colloidal transport properties in porous media under the coupled effects of gravity and hydrodynamics.

by the Science and Technology Development Fund. The funders had no role in study design, data collection and analysis, decision to publish, or preparation of the manuscript.

**Abbreviations:**
**Nomenclature**
A, cross-sectional area of the column, $[L^3]$; $A_1$, $A_2$, $A_3$, defined in (12); b, ratio of the length of the mean free sedimentation section to the radius of the porous medium particle; C, concentration of suspended solids in the liquid phase, $[ML^{-3}]$; $C^*$, mass of particles deposited per unit volume, $[ML^{-3}]$; $C_{out}$, particle concentration at the outlet, $[ML^{-3}]$; $C_c$, curvature coefficient; $C_u$, nonuniformity coefficient; D, hydrodynamic dispersion coefficient, $[L^2T^{-1}]$; $D_{50}$, median particle size, $[L]$; $d_p$, suspended particle diameter, $[L]$; $f_s$, particle settlement correction factor considering the porous medium; g, gravitational acceleration, $[LT^{-2}]$; $g(\cdot)$, injected suspended particle concentration function; $I_0$, modified Bessel functions of the first kind of zeroth order; $I_1$, modified Bessel functions of the first kind of first order; I, particle source intensity, $[MTL^{-3}]$; $k_{at}$, particle adsorption coefficient, $[T^{-1}]$; $k_{de}$, particle release coefficient, $[T^{-1}]$; m, injected particle mass, $[L]$; n, the porosity; Q, flow rate, $[L^3T^{-1}]$; r, Fourier transform variable with respect to spatial coordinate z; R, deposition rate; s, Laplace transform variable with respect to time; $t'$, time to inject particles, $[T]$; t, time, $[T]$; U, average flow velocity of the pore fluid, $[LT^{-1}]$; $U_{tot}$, effective velocity of particles considering gravity, $[LT^{-1}]$; $U_{s(z)}$, z-direction component of the settling velocity of "restricted particles" suitable for granular porous media, $[LT^{-1}]$; v, Darcy flow rate, $[LT^{-1}]$; $V_P$, pore volume of the entire soil column, $[L^3]$; z, spatial coordinate parallel to the flow direction, $[L]$;

**Greek letters**
α, arbitrary constant; $\alpha$,$\alpha_1$,$\alpha_2$, defined in (A14); $\alpha_d$, degree of dispersion, $[L]$; β, flow direction an angle; $\delta(\cdot)$, Dirac delta function; $\rho_p$, suspended particle density, $[M/L^3]$; $\rho_w$, fluid density, $[M/L^3]$; ε, empirical correction factor generated by the influence of the surface of the porous medium particle; τ, dummy variable; $\leq_w$, hydrodynamic viscosity coefficient, $[ML^{-1}T^{-1}]$.

# 1 Introduction

Colloidal particles in groundwater are important carriers of pollutant migration in groundwater-saturated formations due to their strong adsorption properties and mobility. Studies have shown that colloidal particles in groundwater can significantly increase the migration speed and pollutant range, for arsenate [1], heavy metal elements [2], and radionuclides [3]. Therefore, the migration characteristics of colloidal particles in groundwater-saturated strata are of great significance for the study of pollutant migration in a subsurface environment.

A schematic for attached and suspended colloidal particles in the porous space is demonstrated in Fig 1, showcasing the particle statuses of attachment, detachment, suspension. Particles in a saturated porous medium are affected by a variety of forces, including the electrostatic force, the van der Waals force, the hydrodynamic force, and gravity [4, 5]. The factors that lead to particle migration in saturated porous media are quite complex. Various factors that affect the migration of colloids in saturated porous media have been studied experimentally, including the particle size ratio of particles to the porous media [6–8], the fluid velocity between pores [9–12], and the ionic strength [13, 14]. However, there are few studies of the effect of gravity on the migration of colloidal particles in saturated porous media. The effect of the flow direction on particle migration in porous media has not been paid enough attention. However, indoor simulated aquifer tests generally use a water flow orthogonal to the direction of gravity [15, 16], while packed column tests use a flow orthogonal to the direction of gravity [17, 18], a flow opposite to the direction of gravity (upward flow) [19–21], or a flow in the same direction as gravity (downward flow) [22, 23]. Experimental data published in the literature are often compared to classical colloidal filtration theory (CFT) without careful consideration of the experimental flow direction. In addition, Martin Brown [24] found that Pipe inclination (horizontal to vertical) has great influence on particle transportation,and pipe with different deviations could lead to novel evaluations. Therefore, it is necessary to conduct in-depth research on the gravitational effect on particle migration under different seepage directions.

Some studies have considered the effect of gravity on particle migration and deposition in porous media. Wan [25] developed a predictive model for colloidal gravity sedimentation in porous media under hydrostatic conditions and proposed that gravity sedimentation is an important mechanism for bacterial migration in subsurface environments. Constantinos V. Chrysikopoulos [26] studied the effect of the seepage direction on colloid transport in a water-saturated column filled with glass beads at a single flow rate and found that the particle deposition rate was greater when flowing upward than when flowing downward. This result indicated that gravity is an important driving force of the colloidal deposition. force. Chen [27]studied the effect of gravity on the transport characteristics of particles in saturated porous media at different flow rates through soil column tests in two seepage directions: top-down and bottom-up. He found that the lower the seepage velocity, the stronger the effect of gravity. Sharma et al. [28] studied the migration characteristics of particles with different densities in saturated porous media under different centrifugal accelerations through a centrifuge. Yiantsios et al. [29] believed that a larger hydrodynamic force can not only enhance the migration of particles, but also prevent sedimentation due to gravity. It can be seen that the existing research believes that particles in a porous medium have a constant additional sedimentation velocity in the direction of gravity, and this additional sedimentation velocity will affect the migration and deposition characteristics of the particles. Therefore, it is necessary to establish a particle migration model in porous media that considers the effect of gravity and conduct in-depth research on the role of gravity in particle transport under different seepage velocities.

Although there are numerous mathematical models available that describe colloid transport in porous media, in this study, the frequently employed continuum approach was adopted. The phenomenological colloid transport model developed by Massei [30] was extended to

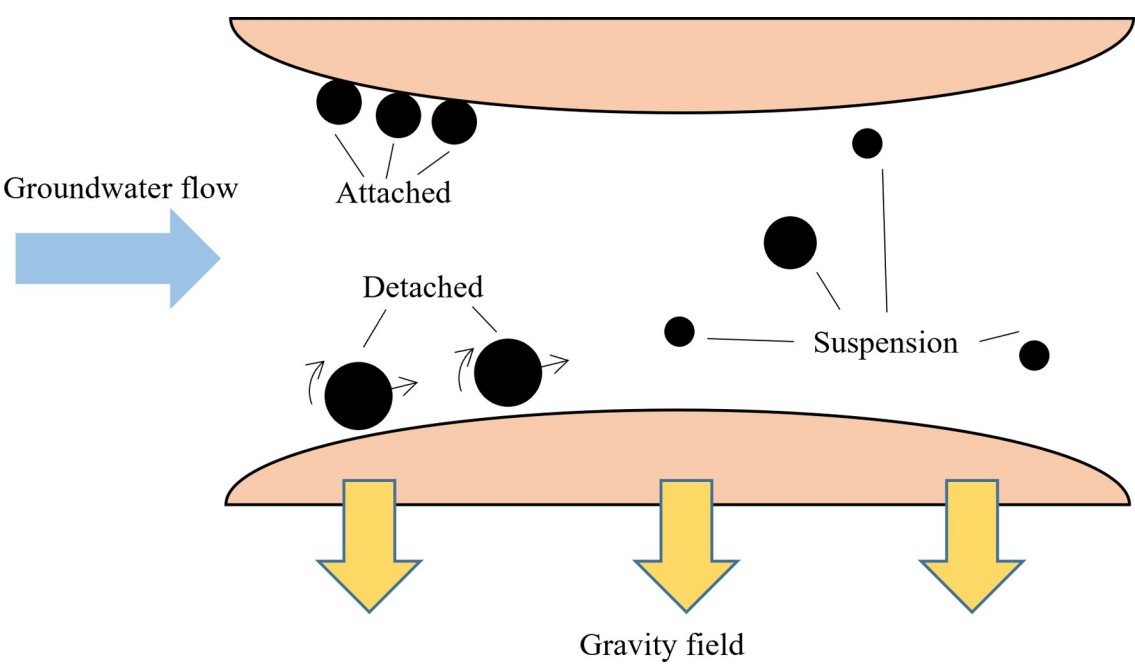

**Fig 1. Schematic of cplloids attachment and suspension.**

account for colloid sedimentation, in order to examine how the flow direction influences colloid fate and transport in porous media. The conventional analytical solution based on a classical convection dispersion model [30–32], in which the influence of the release effect on the deposition kinetics is not considered. Chen and Bai [33] used an analytical solution considering the effect of dispersive flux on deposition kinetics (neglecting detachment) to simulate experimental results. These models are usually used to describe the breakthrough curves (BTCs) of column tests for the case of the so-called "pulse injection", in which the duration of the injected pulse is sufficiently short to be considered instantaneous [30, 32]. However, a potential drawback of these theoretical models is that there is an increasing deviation of the model prediction from the experimental data as the injection time increases. It is very important to establish a solution considering the detachment effect of sorbed particles due to the long release process of particles for a sustained injection.

The steps of this work shows in Fig 2. First, the theoretical model is presented. Based on the classical particle migration model, this paper establishes a particle migration model in saturated porous media that considers the effect of gravity and obtains an analytical solution of particle migration when the particle injection intensity changes with time. Then, a glass bead-packed column experiment was conducted in three seepage directions (horizontal seepage, upward seepage, and downward seepage). The solution containing particles was injected at one end of a column, and the breakthrough curves of the particles were tested at the other end. Finally, A regression analysis was conducted on the test results to deeply study the effect the gravity mechanism on the particle migration characteristics in porous media.

## 2 Theoretical model

### 2.1 Establishment of the governing equations

The one-dimensional transport process of suspended particles in saturated homogeneous porous media considering hydrodynamic dispersion, non-equilibrium adsorption, and

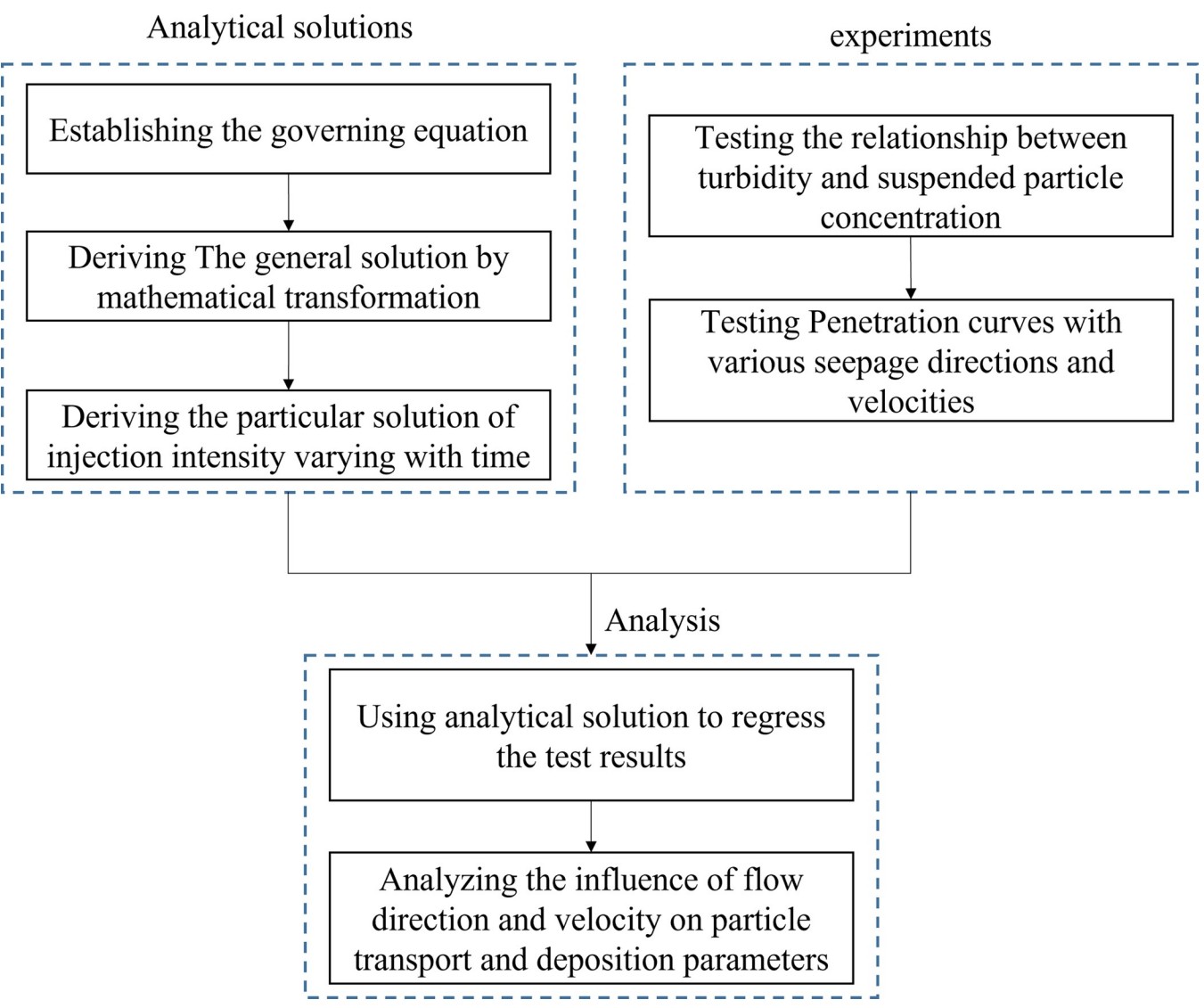

**Fig 2. Flowchart of the steps of the work.**

gravitational effects is governed by the following partial differential equation:

$$\frac{\partial C(z,t)}{\partial t} = D\frac{\partial^2 C(z,t)}{\partial z^2} - U_{\text{tot}}\frac{\partial C(z,t)}{\partial z} - \frac{1}{n}\frac{\partial C^*(z,t)}{\partial t}, \tag{1}$$

$$\frac{1}{n}\frac{\partial C^*(z,t)}{\partial t} = k_{at}C(z,t) - k_{de}\frac{1}{n}C^*(z,t), \tag{2}$$

where $C$ is the concentration of suspended solids in the liquid phase, $[\text{ML}^{-3}]$; $t$ is the time, $[\text{T}]$; $z$ is the spatial coordinate parallel to the flow direction, $[\text{L}]$; $D$ is the hydrodynamic dispersion coefficient, $[\text{L}^2\text{T}^{-1}]$; $n$ is the porosity; $C^*$ is the mass of particles deposited per unit volume, $[\text{ML}^{-3}]$; $k_{at}$ is the particle adsorption coefficient, $[\text{T}^{-1}]$; $k_{de}$ is the particle release coefficient, $[\text{T}^{-1}]$; and $U_{\text{tot}}$ is effective velocity of particles considering gravity, $[\text{LT}^{-1}]$:

$$U_{\text{tot}} = U + U_{s(z)}. \tag{3}$$

where $U$ is the average flow velocity of the pore fluid; and $U_{s(z)}$ is the $z$-direction component of the settling velocity of "restricted particles" suitable for granular porous media and is obtained by correcting the free settling velocity of particles in a static fluid:

$$U_{s(z)} = -f_s \frac{(\rho_p - \rho_w)d_p^2}{18\mu_w} g \cdot \sin\beta, \tag{4}$$

where $\rho_p$ is the suspended particle density, $[M/L^3]$; $\rho_w$ is the fluid density, $[M/L^3]$; $\mu_w$ is the hydrodynamic viscosity coefficient, $[ML^{-1}T^{-1}]$; $d_p$ is the suspended particle diameter, $[L]$; $g$ is the gravitational acceleration, $[LT^{-2}]$; and $\beta$ is the flow direction an angle $(-90° \leq \beta \leq 90°)$ (Fig 3); and $f_s$ is the particle settlement correction factor considering the porous medium [25]:

$$f_s = \frac{b + 0.67}{b + 0.93/\varepsilon}, \tag{5}$$

where $b$ ($b \approx 1$) is the ratio of the length of the mean free sedimentation section to the radius of the porous medium particle; and $\varepsilon$ is the empirical correction factor generated by the influence of the surface of the porous medium particle. When the porous medium is composed of smooth spherical particles, the porous particles only contribute to the tortuosity and do not provide additional frictional resistance, and $f_s \approx 0.9$.

Assuming that there are no adsorbed particles and suspended particles in the porous medium initially, the suspended particles with a time-varying concentration are injected from one end of the column, and the initial and boundary conditions are established as: $C^*(z,0) = C(z,0) = 0$, $C(0,t) = g(t)$, and $C(+\infty,t) = 0$.

Considering that colloidal particles may not be able to penetrate porous media in extreme cases, the colloidal migration model proposed in this paper should have limited requirements for the particle size of porous media. Referring to literature [31] and [34], the particle size ratio of fine particles to porous media is limited to less than 0.1.

## 2.2 Mathematical transformation solution

First, the solution of Eq (2) satisfying the initial conditions was obtained using the Laplace transform as:

$$C^*(z,t) = k_{at} \cdot n \cdot \int_0^t C(z,\tau) \cdot \exp[-k_{de}(t-\tau)]d\tau. \tag{6}$$

After derivation of Eq (6) for $t$, it was inserted into Eq (1) to obtain the expression for the concentration of suspended solids in the liquid phase:

$$\frac{\partial C(z,t)}{\partial t} = D\frac{\partial^2 C(z,t)}{\partial z^2} - U_{tot}\frac{\partial C(z,t)}{\partial z} + k_{de} \cdot k_{at} \cdot \int_0^t C(z,\tau) \cdot \exp[-k_{de}(t-\tau)]d\tau - k_{at}$$
$$\cdot C(z,t). \tag{7}$$

Eq (7) subject to initial and boundary conditions is solved analytically by straightforward but laborious procedures. Taking Laplace transforms with respect to time variable $t$ and Fourier transforms with respect to space variables $z$ of (7) and subsequently employing the transformed initial and boundary conditions, followed by inverse transformations yields the desired analytical solution (see Appendix A in S1 Appendix):

$$C(z,t) = \exp\left(\frac{U_{tot}z}{2D}\right) \cdot \int_0^t g(\tau) \cdot \left[k_{de} \cdot W(z,t-\tau) + \frac{\partial W(z,t-\tau)}{\partial t}\right]d\tau, \tag{8}$$

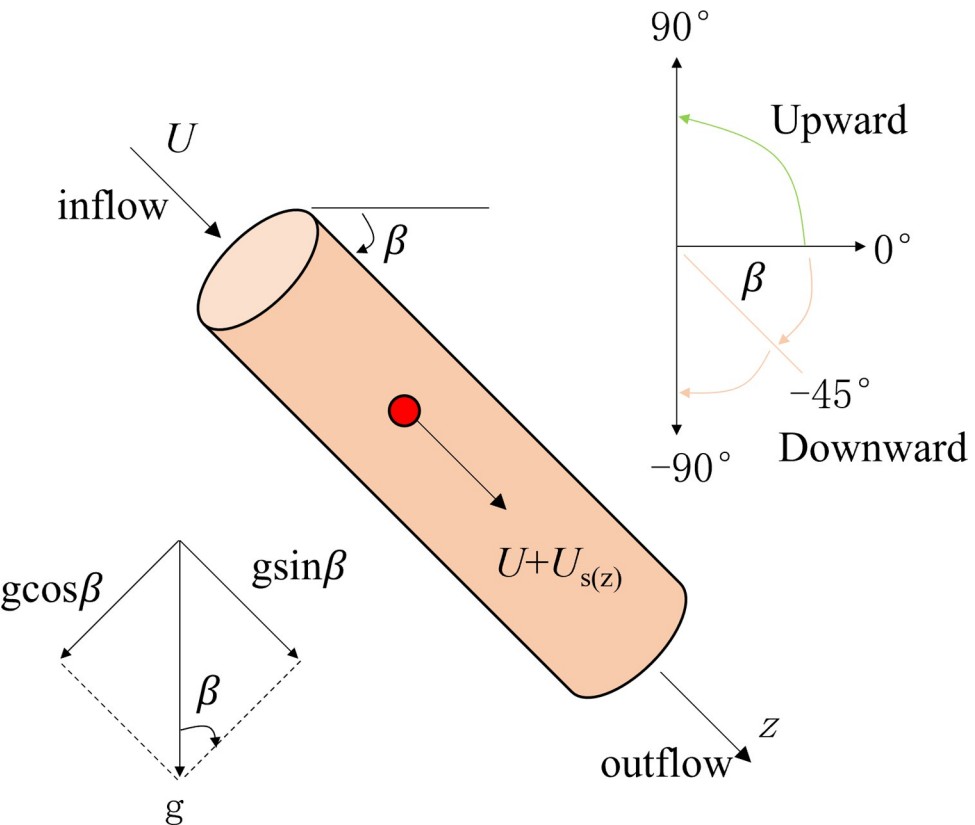

**Fig 3. Conceptual diagram of the particle migration in porous media considering gravity.**

where $\tau$ is a dummy variable and

$$W(z, t) = \exp(-k_{de}t) \int_0^t I_0[2(\alpha\eta(t - \eta))^{1/2}] \cdot \frac{z}{2\eta\sqrt{\pi D\eta}}$$

$$\cdot \exp\left[-\frac{z^2}{4D\eta} - \frac{U_{\text{tot}}^2\eta}{4D} + (k_{de} - k_{at})\eta\right] d\eta, \tag{9}$$

$$\frac{\partial W(z, t)}{\partial t} = \exp(-k_{de}t) \int_0^t \left\{\left(\frac{\alpha\eta}{t - \eta}\right)^{1/2} I_1\left[2(\alpha\eta(t - \eta))^{1/2}\right] - k_{de}I_0\left[2(\alpha\eta(t - \eta))^{1/2}\right]\right\}$$

$$\cdot \frac{z}{2\eta\sqrt{\pi D\eta}} \cdot \exp\left[-\frac{z^2}{4D\eta} - \frac{U_{\text{tot}}^2\eta}{4D} + (k_{de} - k_{at})\eta\right] d\eta + \tag{10}$$

$$\exp^{-k_{de}t} \cdot \frac{z}{2t\sqrt{\pi Dt}} \cdot \exp\left[-\frac{z^2}{4Dt} - \frac{U_{\text{tot}}^2t}{4D} + (k_{de} - k_{at})t\right]$$

Where $\alpha$ is an arbitrary constant; $I_0$ and $I_1$ are the modified Bessel functions of the first kind of zeroth and first order, respectively. It should be noted that the Bessel function relationships $dI_0[\eta]/d\eta = I_1[\eta]$ and $I_0[0] = 1$ were employed in the derivation of (10).

### 2.3 Particular solutions when the injection intensity varies with time

When the suspended particles are injected instantaneously at one end of the column,

$$g(t) = I \cdot \delta(t - t'), \tag{11}$$

where $I = m/Q = m/(vA)$ is the particle source intensity, [MTL$^{-3}$]; $m$ is the injected particle mass, [M]; $Q$ is the flow rate, [L$^3$T$^{-1}$]; $v = nU$ is the Darcy flow rate, [LT$^{-1}$]; $A$ is the cross-sectional area of the column, [L$^3$]; $\delta(\cdot)$ is the Dirac delta function, and $t'$ is the time to inject particles, [T].

Eq (11) is substituted into Eq (8) and using the screening property of the Dirac delta function, the analytical solution of the particle migration under instantaneous injection can be obtained as:

$$C(z, t - t') = \frac{I}{2\sqrt{\pi D}} \cdot A_1(z, t - t') \cdot \left\{ \int_0^{t-t'} A_2(t - t') \cdot A_3(z, \eta) \mathrm{d}\eta + A_3(z, t - t') \right\}, \quad (12)$$

where $A_1(z, t) = \exp\left(\frac{U_{tot}z}{2D} - k_{de}t\right), A_2(t) = \left(\frac{\alpha\eta}{t-\eta}\right)^{1/2} \cdot I_1\left[2(\alpha\eta(t - \eta))^{1/2}\right], A_3(z, t) = \frac{z}{t^{3/2}} \exp\left[-\frac{z^2}{4Dt} - \frac{U_{tot}^2 t}{4D} + (k_{de} - k_{at})t\right]$.

When $I = 1$, the elementary solution of the particle migration in the case of instantaneous injection is:

$$C_P(z, t - t') = \frac{1}{2\sqrt{\pi D}} \cdot A_1(z, t - t') \cdot \left\{ \int_0^{t-t'} A_2(t - t') \cdot A_3(z, \eta) \mathrm{d}\eta + A_3(z, t - t') \right\}. \quad (13)$$

For the case when the injected suspended particle concentration, g(t), varies with time t, the particle concentration can be obtained by integrating Eq (13) in the time domain:

$$C(z, t) = \int_0^t C_P(z, t - t') \cdot g(t') \cdot \mathrm{d}t'. \quad (14)$$

## 3 Experiments introduction

### 3.1 Experiments device

The schematic diagram of the test device is shown in Fig 4. The test device consisted of six parts: a glass column, a bracket, a water tank, a peristaltic pump, a syringe, and a turbidimeter. The length of the glass column was $L$ = 200 mm, the inner diameter was $d$ = 50 mm, and the ratio of the height of the glass column to the inner diameter was 4, which was similar to the one-dimensional seepage test device [32]. The bracket could flexibly fix the sand column at different angles to achieve a variety of seepage directions. The peristaltic pump pumped high-purity deionized water in the water tank into the glass column at a constant speed. When the seepage was stable, a certain concentration of particles in the syringe was injected into the glass column. A turbidimeter was utilized to test the turbidity of the effluent liquid and convert the turbidity from the relationship between the calibrated effluent turbidity and the particle concentration and conduct an in-depth analysis of the particle concentration.

### 3.2 Experiments materials

The porous material was glass beads with particle sizes of 2 mm, and the solid density was 2.65 g/cm$^3$. Prior to the test, the glass beads were pretreated with 0.1 mol/L H$_2$SO$_4$ for 12 hours to remove impurities on the surface of the glass beads, rinsed with deionized water for several times, and finally dried in an oven at a temperature of 105˚C for 48 hours. The column was filled with glass beads and deionized water in layers at 4-cm intervals to ensure uniform filling and that the water level remained above the top plane of the glass beads to avoid air entrapment. The porosity of the sand column was calculated to be 0.41 based on the mass of glass beads packed into the column.

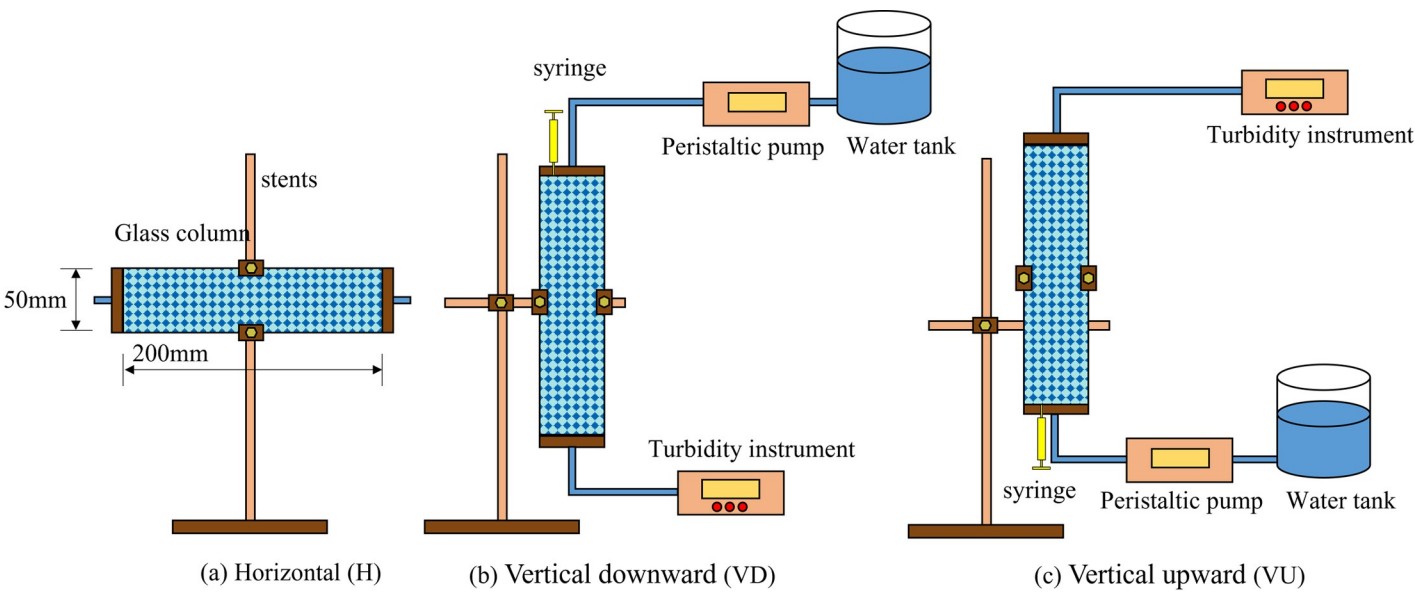

**Fig 4. Schematic diagram of the test device.**

The suspended solid particles were spherical high-purity silica powder, and the particle density was 2.30 g/cm$^3$. The particle size test was conducted using a laser particle size distribution analyzer. Fig 5 shows the gradation curve of the suspended particles. The median particle

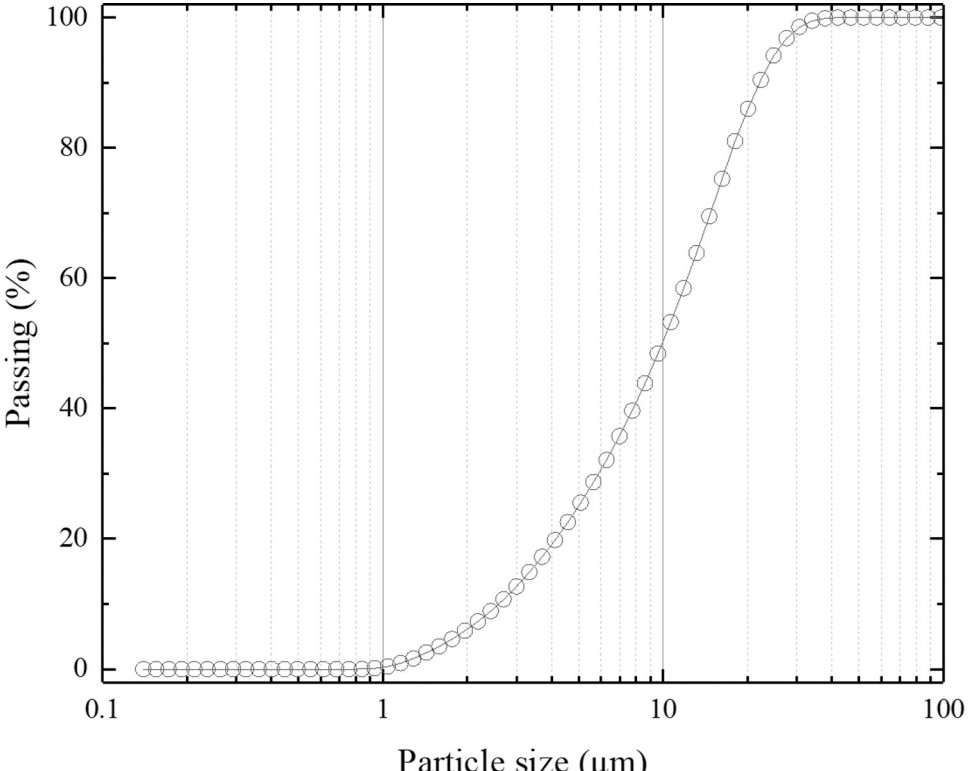

**Fig 5. Suspended particle gradation curve.**

size was $D_{50}$ = 9.94 μm, the curvature coefficient $C_c$ = 1.08, and the nonuniformity coefficient $C_u$ = 4.73. The particle size was relatively concentrated.

## 3.3 Experiments steps

First, the relationship between turbidity and the concentration of suspended particles was established. Based on an initial concentration of 0.4 g/L in 50 ml, 10–50 ml of deionized water was continuously injected to dilute the particle mixture. Immediately after each injection, the turbidity of the mixture was measured using a Hach turbidimeter. The test was considered complete when the turbidity of the mixture was near zero. Each experiment was repeated at least three times, and the mean value was used to minimize the experimental error. A nonlinear regression analysis was performed on the test data ($R^2$ = 0.9995), and the turbidity and concentration of the suspended particles satisfied:

$$Y = 0.00000201X^2 + 0.00294X + 0.00293, \tag{15}$$

where $X$ is the turbidity (NTU); and $Y$ is the concentration (g/L).

During the formal test, deionized water was pumped into the sand column at a constant flow rate to ensure that the turbidity of the effluent liquid was zero, and then 10 mL of the turbid liquid with a concentration of 0.4 g/L was injected into the inlet of the sand column using a syringe. The injection time was very short, less than 1 s. After the injection was complete, the effluent was continuously collected at the column outlet and tested for turbidity. After each test, the glass beads were removed from the sand column, washed with deionized water several times for impurities, and then reloaded into the column. In this experiment, three seepage directions were selected: downward seepage ($\beta$ = −90°), upward seepage ($\beta$ = 90°), and horizontal seepage ($\beta$ = 0°). The seepage velocity ranged from 0.0085 to 0.0255 cm/s, and the Reynolds number ranged from 0.410 to 1.231, which belonged to laminar flow. All of the experiments were conducted at 22–25°C.

# 4 Analysis of the test results

## 4.1 Particle penetration curves under three different seepage directions

To increase the comparability of the experimental results under different flow rates, the ratio of the volume of water flowing through the glass column to the pore volume of the porous medium ($Qt/(nAL)$) was utilized as the horizontal axis and the relative concentration $C_R$ was taken as the vertical axis. The relative concentration $C_R$ is defined as [32]:

$$C_R = C_{out}V_P/m \tag{16}$$

where $C_{out}$ is the particle concentration at the outlet, [ML$^{-3}$]; $V_P$ is the pore volume of the entire soil column, [L$^3$]; $m$ is the injected particle mass, [M].

The particle curves (Fig 6). Eq (12) was used to invert the parameters of the test breakthrough curve. Considering that the particle injection time was very short, and the particle injection amount was small, $k_{de}$ = 0 could be set, and the parameters to be inverted were $U_{tot}$, $D$, and $k_{at}$. The theoretical curve obtained by the fitting is shown in Fig 6. The goodness of fit, $R^2$, of the theoretical curve to the experimental data was all greater than 0.95, indicating that the analytical solution had good applicability.

It can be seen from Fig 6 that for the same seepage velocity, the peaks of the penetration curves in the different seepage directions were ranked vertically downward (VD) > horizontal (H) > vertical upward (VU). The time when the peak of the curve appeared was sorted from small to large as vertical downward (VD) < horizontal (H) < vertical upward (VU). For the

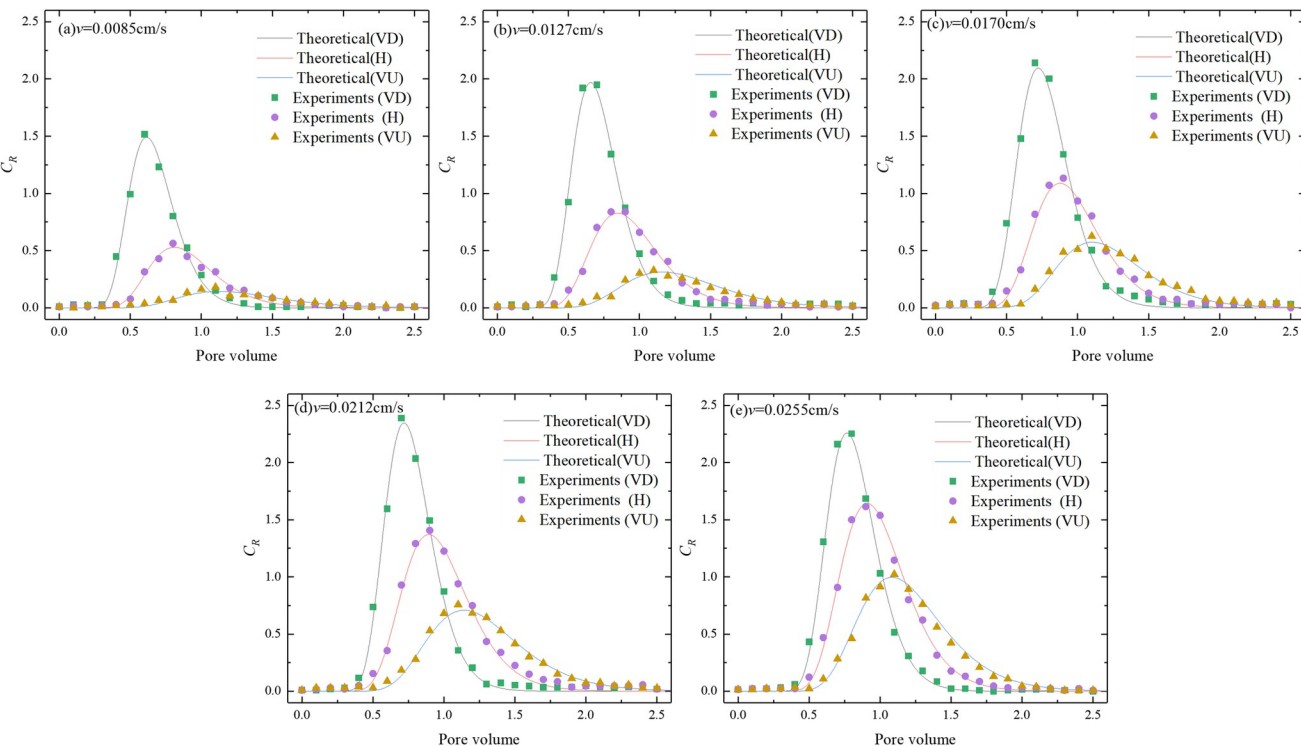

**Fig 6. Particle penetration curves under the different seepage directions and different velocities.**

different seepage velocities, the peak value of the breakthrough curve in the same direction increased with an increase in the seepage velocity. When the seepage velocities were 0.0085, 0.0127, 0.0170, 0.0212, and 0.0255 cm/s, the relative deviations of the peak values of the penetration curves corresponding to the vertical downward (VD) and vertical upward (VU) were 165%, 145%, 114%, 107%, and 78%, respectively, and the relative deviations of the time when the peaks of the penetration curves corresponding to the vertical downward (VD) and vertical upward (VU) appeared were 59%, 55%, 42%, 46%, and 34%, respectively. The analysis showed that with an increase in the seepage velocity, the hydrodynamic force on the particles also increased, and the effect of gravity on the particles was relatively small.

## 4.2 Influence of the seepage direction and seepage velocity on the effective particle velocity

Fig 7 shows the relationship between the effective velocity of the particles and the seepage velocity of the seepage velocity under the conditions of the three seepage directions. The effective velocity is processed dimensionless, and the ordinate becomes $(U_{tot}-U)/U$. The test point was obtained using a parameter inversion of the particle penetration curve using Eq (12). $U$ is the average flow velocity of the pore fluid The average flow rate of the pore fluid was $U = v/n$. The theoretical limit value and the theoretical lower limit value were calculated using Eqs (3) and (4), respectively, corresponding to the seepage direction vertically downward (VD) and vertical upward (VU). In the equation: $d_p = D_{50} = 9.94$ μm, $f = 0.9$, and $\mu_w = 1.01 \times 10^{-3} Pa \cdot s$.

It can be seen from Fig 7 that the experimental inversion and theoretical calculation results both show that the effective velocity of particles under different seepage directions is from large to small vertical downward (VD) > horizontal (H) > vertical upward (VU). The

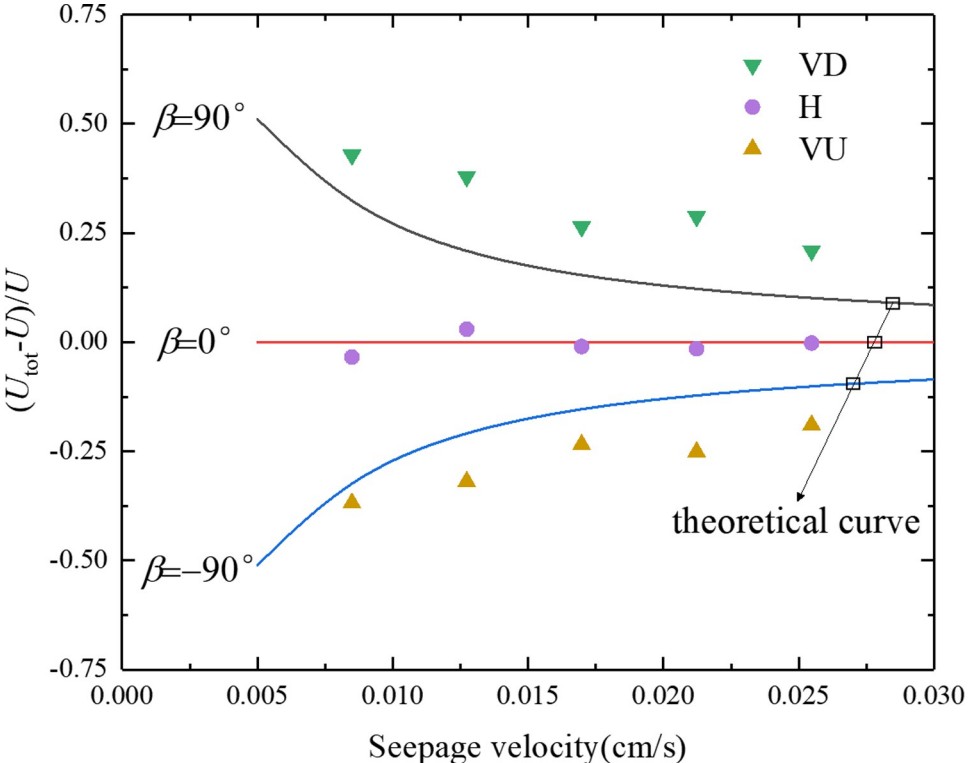

**Fig 7. Relationship between the particle effective velocity and seepage velocity under three different seepage directions.**

experimental inversion value of particle effective velocity is close to the average flow velocity of pore fluid under the condition of seepage direction horizontal (H), and the experimental inversion value of particle effective velocity and pore fluid under the condition of seepage direction vertical downward (VD) and vertical upward (VU) The average flow velocity is quite different, and the deviation direction is consistent with the theoretical and lower limit deviation direction, indicating that the consideration of gravity effect in the particle migration model proposed in this paper is reasonable. The experimental inversion values of the effective particle velocity under the conditions of a vertically downward (VD) and vertically upward (VU) seepage directions exceeded the upper and lower limits of the theoretical calculation, which may have been because the suspended particles used in the test had a particle size range, not a single particle size, which would make the theoretical calculation results and experimental results have a certain difference.

## 4.3 Influence of the seepage direction and seepage velocity on the particle dispersion characteristics

Dispersion is an important mechanism for particle migration in porous media. The dispersion characteristics of particles in porous media are often characterized by the degree of dispersion that is defined as [32]: $\alpha_d = D/U$.

Fig 8 shows the relationship between the dispersion of porous media and the seepage velocity under three different seepage directions. It can be seen from Fig 8 that with an increase in the seepage velocity, the dispersion displayed a downward trend. When the seepage direction was vertically downward (VD), the dispersion decreased from 0.96 to 0.60. When the seepage

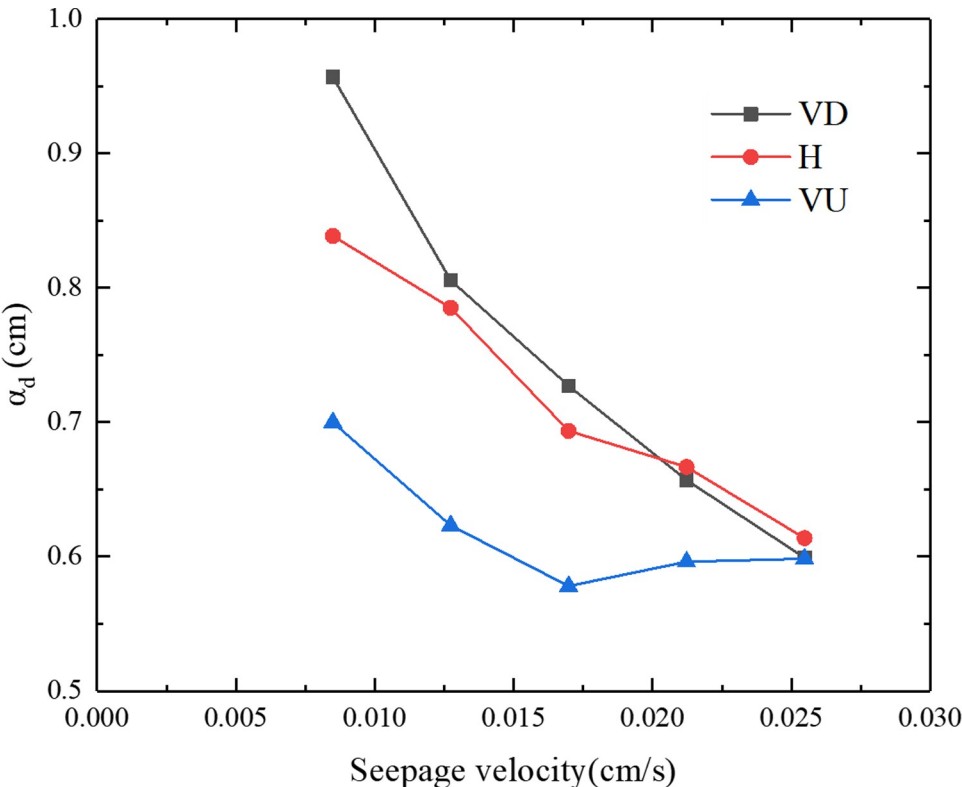

**Fig 8. The relationship between the dispersion and seepage velocity of the porous media under three different seepage directions.**

direction was horizontal (H), the dispersion decreased from 0.84 to 0.61. When the seepage direction was vertically upward (VU), the dispersion decreased from 0.70 to 0.60. When the direction of seepage was opposite to the direction of gravity, the effect of the seepage velocity on dispersion was minimal. When the seepage velocities were 0.0085, 0.0127, 0.0170, 0.0212, and 0.0255 cm/s, the relative deviations in the dispersion corresponding to the vertically downward (VD) and the vertically upward (VU) were 31%, 26%, 23%, 9.7%, and 0.1%, and the smaller the seepage velocity, the greater the dispersion difference in the different seepage directions.

## 4.4 Influence of the seepage direction and seepage velocity on the particle deposition characteristics

The deposition rate, $R$, was used to evaluate the degree of particle deposition. The deposition rate was defined as the ratio of the mass of particles deposited on the porous medium to the total mass of injected particles. The calculation equation is [32]:

$$R = 1 - \int_0^t QC(t)\mathrm{d}t/m \tag{17}$$

where C(t) is the theoretical value of the particle concentration in the effluent of the glass column that is obtained by fitting the test point with Eq (12).

Fig 9 shows the relationship between the particle deposition rate and the seepage velocity under three different seepage directions. It can be seen from Fig 9 that under the same seepage direction, the particle deposition rate decreased with an increase in the seepage velocity. For different seepage directions, the order of the particle deposition rate from large to small was

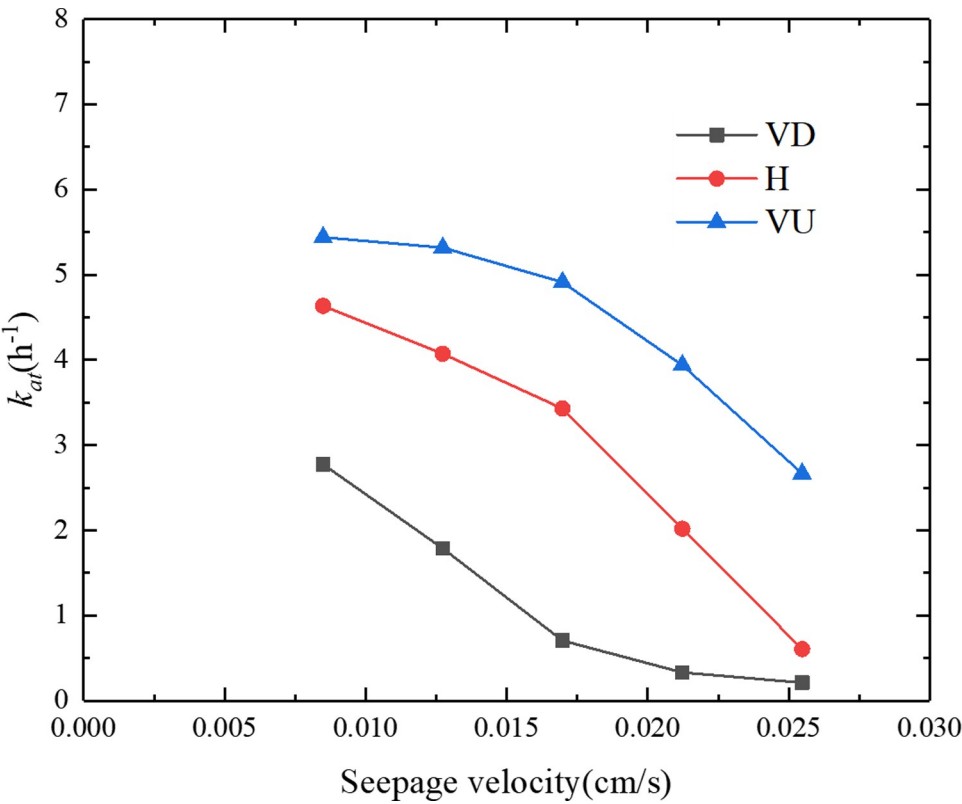

**Fig 9. Relationship between the particle deposition rate and the seepage velocity under the three different seepage directions.**

vertically upward (VU)>horizontal (H)>vertically downward (VD). When the seepage direction was vertically downward (VD), the deposition rate decreased from 40.1% to 1.6%. When the seepage direction was horizontal (H), the deposition rate decreased from 69.4% to 5.3%. When the seepage direction was vertically upward (VU), the deposition rate dropped from 87.4% to 25.3%. Fig 10 shows the relationship between the particle adsorption coefficient and the seepage velocity under the three different seepage directions. It can be seen from Fig 10 that under the same seepage direction, the particle adsorption coefficient decreased with an increase in the seepage velocity. For the different seepage directions, the order of particle adsorption coefficient from large to small was vertically upward (VU) > horizontal (H) > vertically downward (VD). When the seepage direction was vertically downward (VD), the particle adsorption coefficient dropped from 2.78 h$^{-1}$ to 0.21 h$^{-1}$. When the seepage direction was horizontal (H), the particle adsorption coefficient dropped from 4.63 h$^{-1}$ to 0.61 h$^{-1}$. When the seepage direction was vertically upward (VU), the particle adsorption coefficient decreased from 5.44 h$^{-1}$ to 2.67 h$^{-1}$. The analysis demonstrated that the greater the seepage velocity, the greater the hydrodynamic force on the particles, and the less conducive to particle deposition. When the seepage direction was opposite to the direction of gravity, gravity counteracted part of the hydrodynamic effect, making it easier for particles to deposit in the porous medium.

## 5 Summary and conclusions

In this paper, a mathematical model of particle migration in a saturated porous medium that considered the effect of gravity and the release effect on the deposition dynamics, and the

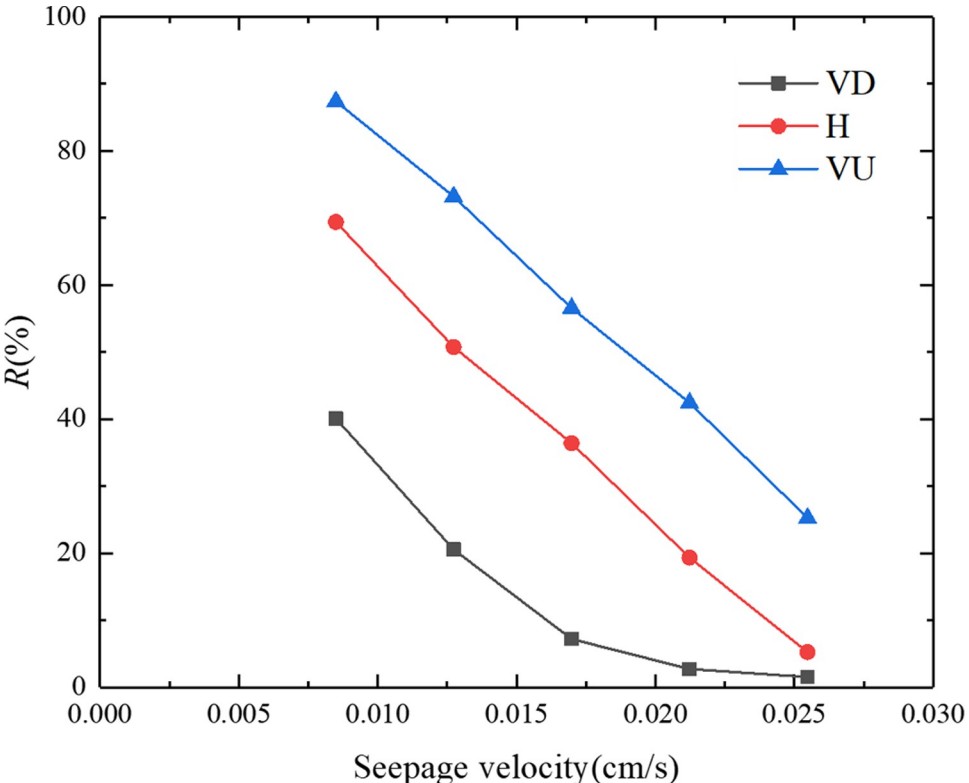

**Fig 10. Relationship between the particle adsorption coefficient and the seepage velocity under the three different seepage directions.**

analytical solution to the particle migration problem when the particle injection intensity varied with time was obtained using an integral transformation. The correctness and rationality of the obtained analytical solution were verified using a comparative analysis of the experimental and theoretical results of the particle migration under the point source instantaneous injection method. The effects of the seepage velocity on the particle migration parameters in three seepage directions were analyzed, and the following conclusions were drawn:

(1) Under the same seepage direction, the peak value of the breakthrough curve increased with an increase in the seepage velocity. Under the condition of the same seepage velocity, the peaks of the breakthrough curves in the different seepage directions were ranked vertically downward (VD)>horizontal (H)>vertically upward (VU), and the peak times of the breakthrough curves appeared from small to small. The big order was vertically down (VD)<horizontal (H)<vertically up (VU). The smaller the seepage velocity, the greater the relative deviation between the peaks of the breakthrough curves in the different seepage directions and the time when the peaks of the breakthrough curves appeared and the more obvious the gravitational effect.

(2) Gravity and the seepage velocity jointly affected the particle dispersion characteristics. Under the condition of the same seepage direction, the dispersity decreased with an increase in the seepage velocity. When the seepage direction was opposite to the direction of gravity, the effect of the seepage velocity on dispersibility was the least. Under the condition of the same seepage velocity, the order of dispersity from large to small was vertically downward (VD)>horizontal (H)>vertically upward (VU), and the smaller the seepage velocity, the greater the difference in the dispersity in the different seepage directions.

(3) Under the condition of the same seepage direction, the adsorption coefficient and deposition rate of the particles decreased with an increase in the seepage velocity, and larger hydrodynamic conditions were not conducive to particle deposition. Under the same seepage velocity, the adsorption coefficient and deposition rate of the particles were ranked vertically upward (VU)>horizontal (H)>vertically downward (VD). It can be seen that gravity was an important mechanism for particle migration in saturated porous media. The larger the particle size and density, the smaller the seepage velocity, and the greater the difference in the results of the particle migration under the different seepage directions.

When the surface of porous media particles is irregular or rough, it is difficult to accurately calculate the particle settlement correction factor in the theoretical model. Therefore, smooth glass spheres are selected as the porous media in this paper, while the actual porous media surface is irregular or rough. Future studies should consider the role of gravitational force on colloid transport in saturated as well as unsaturated packed columns with irregular or coarse porous medium particles.

## Supporting information

**S1 Appendix.**
(DOCX)

**S1 Data.**
(XLSX)

## Acknowledgments

We thank LetPub (www.letpub.com) for its linguistic assistance during the preparation of this manuscript.

## Author Contributions

**Conceptualization:** Xiaoming Liu.

**Data curation:** Shujie Tu.

**Software:** Hongjiang Cai.

**Validation:** Shujie Tu.

**Writing – original draft:** Shujie Tu.

**Writing – review & editing:** Xiaoming Liu.

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
