## [Decision Letter · Decision Letter 0]

30 Aug 2022

PONE-D-22-19180Effect of gravity on particle transport in a saturated porous medium: Analytical solutions and experimentsPLOS ONE

Dear Dr. Xiaoming,

Thank you for submitting your manuscript to PLOS ONE. After careful consideration, we feel that it has merit but does not fully meet PLOS ONE’s publication criteria as it currently stands. Therefore, we invite you to submit a revised version of the manuscript that addresses the points raised during the review process.

ACADEMIC EDITOR: Both reviewers suggested for major revision for the manuscript. It is essential to satisfy the reviewers. 

We look forward to receiving your revised manuscript.

Kind regards,

Ajaya Bhattarai

Academic Editor

PLOS ONE

Journal Requirements:

4. Please ensure that you include a title page within your main document. You should list all authors and all affiliations as per our author instructions and clearly indicate the corresponding author.

Additional Editor Comments:

Both reviewers suggested for major revision.

Reviewers' comments:

Reviewer's Responses to Questions

**Comments to the Author**

1. Is the manuscript technically sound, and do the data support the conclusions?

Reviewer #1: Yes

Reviewer #2: Partly

2. Has the statistical analysis been performed appropriately and rigorously? 

Reviewer #1: N/A

Reviewer #2: N/A

3. Have the authors made all data underlying the findings in their manuscript fully available?

Reviewer #1: No

Reviewer #2: Yes

4. Is the manuscript presented in an intelligible fashion and written in standard English?

Reviewer #1: Yes

Reviewer #2: Yes

5. Review Comments to the Author

Reviewer #1: Review of:

[Manuscript PONE-D-22-19180

Title: Effect of gravity on particle transport in a saturated porous medium: Analytical solutions and experiments

Authors: ---]

I have read the paper with interest.

The manuscript topic is of interest for “PLOS ONE”.

The authors offer a work that potentially can improve our knowledge about the topic of interest. I suggest that the authors consider a revision of their work along with the following suggestions and questions.

Reviewer #2: This paper presents analytical developments on transient one-dimensional colloid transport in porous media considering the effect of gravity, and backs them up with experiments that show a good agreement with the theoretical solution developed. The paper deals with a topic of interest for PLOS ONE readership. In the revised version, the authors should consider mentioning in the title that colloids are considered, or, almost equivalently, that the 1-D ADE including sorption is considered and solved, otherwise without sorption the mathematical problem would be trivial. The analytical solution by means of the Laplace transport is clever and seems original, however I would like to see in the revised version a clear indication of what is original and what was earlier achieved in the earlier works on the solution of the ADE with sorption that are mentioned in the introduction. A third item clearly needing revision is the sensitivity analysis, which is lengthy and unclear: the authors should consider rewriting their analytical solution and all relevant outputs in dimensionless form (this is easily achieved by finding adequate scales, and is now standard in this type of papers), and performing the comparison with experiments in such form. Discussing the comparison between theory and experiments in dimensionless form achieves greater generality. Moreover, the manuscript requires substantial rewriting and editing to improve the grammar, syntax and English usage. This, in itself, is a pressing issue to ensure publication.

6. PLOS authors have the option to publish the peer review history of their article (what does this mean?). If published, this will include your full peer review and any attached files.

Reviewer #1: No

Reviewer #2: No

---

## [Author Response · Author response to Decision Letter 0]

8 Sep 2022

Dear Editors and Reviewers:

Thank you for your letter and for the reviewers’ comments concerning our manuscript entitled “Effect of gravity on particle transport in a saturated porous medium: Analytical solutions and experiments” (PONE-D-22-19180). Those comments are all valuable and very helpful for revising and improving our paper, as well as the important guiding significance to our researches. We have studied comments carefully and have made correction which we hope meet with approval. Revised portion are marked in red in the paper. The main corrections in the paper and the responds to the reviewer’s comments can be found in the document "Response to Reviewers".

---

## [Editor Report · Decision Letter 1]

22 Sep 2022

Effect of gravity on colloidal particle transport in a saturated porous medium: Analytical solutions and experiments

PONE-D-22-19180R1

Dear Dr. Xiaoming,

We’re pleased to inform you that your manuscript has been judged scientifically suitable for publication and will be formally accepted for publication once it meets all outstanding technical requirements.

Kind regards,

Ajaya Bhattarai

Academic Editor

PLOS ONE
---

## [Editor Report · Acceptance letter]

26 Sep 2022

PONE-D-22-19180R1 

Effect of gravity on colloidal particle transport in a saturated porous medium: Analytical solutions and experiments 

Dear Dr. Liu:

I'm pleased to inform you that your manuscript has been deemed suitable for publication in PLOS ONE. Congratulations! Your manuscript is now with our production department. 

Kind regards, 

on behalf of

Dr. Ajaya Bhattarai 

Academic Editor

PLOS ONE